TOPICAL REVIEW

# The neurorehabilitation of post-stroke dysphagia: Physiology and pathophysiology

Ayodele Sasegbon[1] (iD), Ivy Cheng[1,2,3] and Shaheen Hamdy[1] (iD)

[1]*Division of Diabetes, Endocrinology and Gastroenterology, School of Medical Sciences, Centre for Gastrointestinal Sciences, Faculty of Biology, Medicine and Health, Salford Royal Foundation Trust, University of Manchester, Manchester, UK*
[2]*Academic Unit of Human Communication, Learning, and Development, Faculty of Education, The University of Hong Kong, Hong Kong, China*
[3]*Institute for Biomagnetism and Biosignalanalysis, University of Münster, Münster, Germany*

Handling Editors: Laura Bennet & Richard Carson

The peer review history is available in the Supporting Information section of this article (https://doi.org/10.1113/JP285564#support-information-section).

The Journal of Physiology

**Abstract**  Swallowing is a complex process involving the precise contractions of numerous muscles of the head and neck, which act to process and shepherd ingested material from the

**Ayodele Sasegbon** is a gastroenterology academic clinical lecturer at the University of Manchester. His research interests lie in the fields of nutrition, gastrointestinal physiology, dysphagia and neuromodulation. Despite how common dysphagia is in the general population, it has long been sidelined by other forms of pathology. He works with a large team of people trying to cast light on this comparatively under-studied area of physiology. **Ivy Cheng** is an Assistant Professor of Speech-Language Pathology at the University of Hong Kong and an honorary postdoctoral research associate at the University of Manchester. Her areas of expertise lie in swallowing neurophysiology and dysphagia, and her primary research focuses on utilising neurostimulation techniques to enhance neuroplasticity in dysphagia rehabilitation. In recognition of her research excellence, she was awarded the Women in Research (WiRe) Fellowship at the University of Münster in Germany in 2023. **Shaheen Hamdy** is currently Professor of Neurogastroenterology based in the Centre for GI Sciences, within the Division of Diabetes, Endocrinology and Gastroenterology, Faculty of Biology, Medicine and Health, University of Manchester. His research interests include neural mechanisms within the gastrointestinal system, with a particular focus on neuroplasticity and functional recovery following brain injury using human swallowing as an experimental model.

oral cavity to its eventual destination, the stomach. Over the past five decades, information from animal and human studies has laid bare the complex network of neurones in the brainstem, cortex and cerebellum that are responsible for orchestrating each normal swallow. Amidst this complexity, problems can and often do occur that result in dysphagia, defined as impaired or disordered swallowing. Dysphagia is common, arising from multiple varied disease processes that can affect any of the neuromuscular structures involved in swallowing. Post-stroke dysphagia (PSD) remains the most prevalent and most commonly studied form of dysphagia and, as such, provides an important disease model to assess dysphagia physiology and pathophysiology. In this review, we explore the complex neuroanatomical processes that occur during normal swallowing and PSD. This includes how strokes cause dysphagia, the mechanisms through which natural neuroplastic recovery occurs, current treatments for patients with persistent dysphagia and emerging neuromodulatory treatments.

(Received 2 December 2023; accepted after revision 29 February 2024; first published online 22 March 2024)

**Corresponding author** S. Hamdy: Division of Diabetes, Endocrinology, and Gastroenterology, School of Medical Sciences, Centre for Gastrointestinal Sciences, Faculty of Biology, Medicine and Health, Salford Royal Foundation Trust, University of Manchester, Clinical Sciences Building, Manchester, Eccles Old Road, Salford M6 8HD, UK.    Email: shaheen.hamdy@manchester.ac.uk

**Abstract figure legend** The organisation of the swallowing central pattern generator (CPG) located in the medulla oblongata. The CPG is consisted of two groups of neurones: the dorsal swallowing group (DSG) and the ventral swallowing group (VSG). Neurones in the DSG receive inputs from peripheral receptors and supramedullary structures and activate the VSG neurones. The VSG neurones then send signals to the motor nuclei.

## Introduction

Swallowing is essential for survival in humans. It is a complicated physiological process that relies on the timely co-ordination of oropharyngeal and oesophageal muscles to transport food safely and efficiently from the mouth to the stomach. This process is mediated by the central (CNS) and peripheral nervous systems. When the nervous system is compromised following neuro-logical insults (e.g. after strokes), dysphagia may occur. Dysphagia can lead to devastating impacts on physical and psychosocial well-being. Although some patients with post-stroke dysphagia (PSD) may recover within the first few weeks following a stroke, others may continue to experience difficulties in swallowing. For these patients, neurorehabilitation becomes the key to regaining the ability to swallow safely. Our increased knowledge of the physiology of swallowing and pathophysiology of PSD has served as the foundation for the development of current neurorehabilitation approaches and will aid the creation and further development of novel therapeutic strategies.

Here, we present an overview of the physiology and neurophysiology of swallowing. This is followed by a discussion on the pathophysiology of PSD, summarising evidence from neuroimaging and neurophysiological studies on the mechanisms by which stroke can affect the normal swallowing process. Finally, we discuss the mechanisms underlying functional recovery following stroke and potential management approaches for PSD.

## Anatomy of swallowing

Over 25 pairs of muscles in the jaw, lips, cheek, soft palate, larynx, pharynx and oesophagus are involved in swallowing (Table 1) (Dubner, 2013; Hennessy & Goldenberg, 2016; Sasegbon & Hamdy, 2017; Yamada et al., 2005). Throughout the process of swallowing, these muscles co-ordinate in an organised and dynamic manner to ensure each fluid or food bolus is processed (as required) and transported safely and efficiently from the oral cavity to the stomach. Some muscles, such as the muscles which open and close the jaw, function antagonistically to each other. Other muscles such as the oropharyngeal muscles, may activate simultaneously to perform complex movements during the swallowing reflex, whereas muscles involved in the oesophageal stage may activate sequentially to generate peristaltic waves (Jean, 2001). Table 1 summarises the anatomy of swallowing and the main functions of each muscle group during swallowing.

## The physiology of swallowing

Swallowing involves a stereotyped motor pattern (Doty, 1951; Jean, 2001), which can be classified into three main stages depending on the location of the bolus: oral stage, pharyngeal stage and oesophageal stage.

**Table 1. Anatomy of the oropharyngeal and oesophageal stages of swallowing**

| Muscle groups | | Individual muscles | Main function in swallowing |
|---|---|---|---|
| Jaw | Jaw closing muscles | Masseter<br>Temporalis<br>Medial pterygoid | Mastication and food processing |
| | Jaw opening muscles | Digastric<br>Mylohyoid<br>Suprahyoid<br>Lateral pterygoid | |
| Lips and cheek | | Orbicularis oris<br>Risorius<br>Lip elevators<br>Lip depressors<br>Buccinator | Containment of the food bolus in the oral cavity during the oral stage |
| Tongue | Intrinsic muscles | Vertical<br>Transverse<br>Superior longitudinal<br>Inferior longitudinal | Manipulation of the food bolus in the oral cavity to enable processing and propulsion during the oral stage |
| | Extrinsic muscles | Genioglossus<br>Styloglossus<br>Hyoglossus<br>Palatoglossus | |
| Soft palate | | Levator veli palatine<br>Tensor veli palatine<br>Palatoglossus<br>Palatopharyngeus<br>Musculus uvulae | Elevation of the soft palate during the pharyngeal stage to seal the entrance from the oral to the nasal cavity |
| Larynx | Intrinsic laryngeal muscles | Lateral cricoarytenoid<br>Transverse and oblique arytenoid<br>Aryepiglotticus<br>Thyroepiglotticus | Control of the opening to the laryngeal inlet |
| Pharynx | | Anterior digastric<br>Geniohyoid<br>Stylohyoid<br>Styloglossus | Constriction of the pharyngeal wall and elevation of the pharynx and larynx during the pharyngeal stage |
| | External circular muscles of pharyngeal wall | Superior, middle, and inferior constrictors<br>Palatoglossus | |
| | Internal longitudinal muscles of pharyngeal wall | Palatopharyngeus<br>Stylopharyngeus<br>Salpingopharyngeus | |
| Oesophagus | Upper oesophageal sphincter | Cricopharyngeus | Relaxation of the upper oesophageal sphincter allows passage of the bolus from the pharynx to the oesophagus<br>Propulsion of the bolus towards the stomach through peristaltic waves during the oesophageal stage |
| | | Thyropharyngeus | |
| | Oesophagus | Internal circular muscle<br>External longitudinal muscle | |

The oral stage is voluntary and can be subdivided into oral preparatory and oral propulsive stages (Hennessy & Goldenberg, 2016). The motor events that occur during the oral stage differ between liquids and sold food. Liquids require no preparation or processing once ingested (Sasegbon & Hamdy, 2017). During the propulsive stage, the tongue is pressed against the hard palate and flexed squeezing the liquid bolus backwards towards the pharynx. This process introduces the bolus into the pharynx and initiates the pharyngeal stage of swallowing (Matsuo & Palmer, 2009; Sasegbon & Hamdy, 2017). By contrast, there is an overlap in the timing of the oral preparatory, propulsive and pharyngeal stages during the processing of solid food (Hiiemae & Palmer, 1999; Palmer et al., 1992). There are three substages during the oral stage when it comes to swallowing solid food. The first stage is transport stage I, where food is manipulated in the oral cavity by lateral tongue movements. This is followed by the food processing stage, where food is rendered into a paste, of variable consistency, through mastication and mixing with saliva (Jean, 2001; Matsuo & Palmer, 2009), and transport stage II, which is equivalent to the oral propulsive stage for liquids. Food processing may occur again following transport stage II, depending on if there is food remaining in the oral cavity (Matsuo & Palmer, 2009).

The pharyngeal stage is involuntary and is marked by the contraction of suprahyoid muscles, which constitute part of the 'leading complex' that is responsible for the initiation of swallowing reflex (Jean, 2001). The swallowing reflex is triggered by the pressure from the bolus as it reaches the mucosal regions innervated by the superior laryngeal nerve at the posterior aspect of the oral cavity and pharyngeal entrance during the oral propulsive stage (Miller, 1982). Once triggered, in a normal swallow, an invariable sequence of motor events occurs in an 'all-or-none' manner, irrespective of the bolus characteristics, suggesting the presence of a central pattern generator for swallowing (Ertekin & Aydogdu, 2003).

During the pharyngeal stage, the soft palate elevates as the bolus reaches the pharynx to prevent nasal regurgitation. The bolus is pushed towards the pharyngeal wall, during which the pharyngeal constrictor muscles contract sequentially to squeeze the bolus downwards. Several mechanisms occur to prevent the bolus from entering the airway. The suprahyoid and thyrohyoid muscles contract to move the hyoid bone and larynx superiorly and anteriorly (Matsuo & Palmer, 2009). The epiglottis is tilted backward to narrow the laryngeal vestibule, whereas the laryngeal adductors contract to close the glottis, providing an extra layer of protection of the airway (Miller, 1982). The bolus then passes into the oesophagus following the opening of the upper oesophageal sphincter, which is a reflexive process initiated by the contraction of the suprahyoid and thyro-hyoid muscles, relaxation of the cricopharyngeus and pressure from the bolus (Ertekin & Aydogdu, 2003; Matsuo & Palmer, 2009). The entire oropharyngeal stage takes place in 0.6–1.0 s (Jean, 2001).

The oesophageal stage begins when the bolus passes through the upper oesophageal sphincter. It involves peristaltic contraction of oesophageal muscles from the proximal to distal oesophagus (Jean, 2001). This peristaltic wave pushes the bolus downwards towards the lower oesophageal sphincter, which is normally closed to prevent reflux of gastric contents. The oesophageal stage takes an average of 8 s (Miller, 1982).

## Factors affecting the physiology of swallowing

Although the sequence of motor events during the pharyngeal stage of swallowing is invariable in a normal swallow, the timing and duration of these events may be altered by descending signals from the motor cortex or the brainstem, which may be driven by bolus induced sensory inputs from the oropharyngeal muscles (Miller, 1982). Studies have shown that the viscosity, volume, temperature and chemical properties of a bolus can alter the physiology of swallowing. For example, when swallowing thickened fluids, the oral and pharyngeal transit times and duration of pharyngeal peristaltic waves are longer, and the opening of the upper oesophageal sphincter is prolonged and increased (Dantas et al., 1990). Moreover, the closure of the true vocal folds occurs earlier and for longer during swallowing of thickened fluids compared to thin fluids (Inamoto et al., 2013). An increased bolus volume results in increased bolus transit time, earlier true vocal fold closure and opening of the upper oesophageal sphincter, and an increased cross-sectional area of the upper oesophageal sphincter opening (Bisch et al., 1994; Pongpipatpaiboon et al., 2022). Cold fluids reduce the timing of swallowing events compared to room temperature and hot fluids (Michou et al., 2012). Carbonated fluids lead to better swallowing co-ordination compared to still water (Michou et al., 2012). Furthermore, boluses with a bitter taste increase the latency of an electrically elicited swallowing reflex, whereas the addition of monosodium glutamate counter-acts such delay (Otake et al., 2016).

Ageing is a natural process that involves physiological and neurological changes that may impact physical function. The term 'presbyphagia' has been used to describe ageing-related changes in the physiology of swallowing that may increase the risk of dysphagia but do not necessarily constitute dysphagia. Sarcopenia, which is the loss of skeletal muscle mass, strength and function as a result of ageing, may impact the oropharyngeal stage of swallowing and is an independent risk factor for dysphagia in hospitalised elderly people (Maeda & Akagi, 2016).

Studies have shown that people over 70 years of age have significant atrophy of their geniohyoid muscles regardless of gender, which is associated with aspiration (Feng et al., 2013), reduced tongue strength (Park et al., 2015; Vanderwegen et al., 2013), reduced jaw opening force (Iida et al., 2013), reduced pharyngeal wall thickness and increased pharyngeal lumen area (Molfenter et al., 2015). Apart from sarcopenia, elderly people may experience changes in the perception of taste, smell and consistency of boluses. Studies found that the detection and recognition thresholds of sweet, sour, bitter and savoury tastes and smells were elevated in the elderly, suggesting a reduction in gustatory and olfactory sensitivity (Schiffman, 1979; Stevens & Cain, 1993). Moreover, the oropharyngeal perception of fluid viscosity deteriorates with increasing age (Smith et al., 2006). Other physiological changes, including reduced muscle elasticity, changes in the cervical spine, impaired dental status and reduced saliva production, may also alter the physiology of swallowing (Wirth et al., 2016). These physical and physiological changes may lead to a prolonged oral onset, pharyngeal stage transit time and duration of upper oesophageal sphincter opening, a decreased swallow volume, increased residue, changes in upper oesophageal sphincter pressures, and increased risks of penetration or aspiration (Jardine et al., 2020; Robbins et al., 1992; Rofes et al., 2010). These presbyphagic and sarcopenic changes act to reduce an individuals swallowing reserve and are though to be the reason why conditions which further stress the swallowing motor system such as sepsis, are associated with decompensation and dysphagia in older individuals (Sasegbon et al., 2018).

Apart from bolus characteristics and ageing, the physiology of swallowing may be different between males and females. Studies have found that males have a shorter duration of oropharyngeal transit (Dantas et al., 2009) and upper oesophageal sphincter opening (Robbins et al., 1992), as well as a delayed onset of laryngeal elevation (Endo et al., 2020). These differences in swallowing physiology may be attributed to differences in anatomy and muscle compositions between genders. Finally, there is a small amount of evidence mainly originating from studies with small sample sizes that suggests that there may be an interaction between age and gender on swallowing physiology. Some studies have reported that males exhibit a more significant deterioration in sensitivity of fluid viscosity with increasing age than females (Smith et al., 2006), and that the duration of swallowing cessation (the holding of breath at the point of swallowing) during saliva swallows decreases with increasing age in males but increases in females (Hiss et al., 2001).

## Neurophysiology of swallowing

To understand the pathophysiology of PSD, it is important to appreciate the intricacies of the neurological basis for swallowing. Much of the basic knowledge concerning the neural control of swallowing has traditionally come from studies on anaesthetised or awake mammals using direct electrical stimulation of the brain and extracellular or intracellular recordings of neuronal activity (Cheng, Takahashi, et al., 2022). These studies collectively showed that swallowing can be elicited through stimulating peripheral nerves, the brainstem, and various cortical and sub-cortical structures (Cheng, Takahashi, et al., 2022; Jean, 2001; Miller, 1972).

**Brainstem.** The brainstem is considered to be the command centre for swallowing. It receives afferent inputs from both peripheral nerves and the cerebral cortex, integrates and processes the information, and generates efferent motor signals for the execution of the swallowing motor sequence (Dimant, 1996). Based on microelectrode recordings in animals, studies have identified that swallowing neurones, including motor neurones and interneurones, are located in the medulla oblongata, which is at the base of the brainstem (Jean, 2001). These neurones form a complex unit termed the swallowing central pattern generator (CPG), which is responsible for the formation and regulation of the motor sequence of swallowing (Jean, 2001) (see Graphical abstract). Neurones within the CPG are classified into two groups: the dorsal swallowing group (DSG), located within the nucleus tractus solitarii and the adjacent reticular formation, and the ventral swallowing group (VSG), located in the ventrolateral medulla (Jean, 2001).

The DSG comprises generator interneurones that are involved in the triggering and sequencing of motor events (Jean, 2001). These neurones exhibit a pattern of discharges before swallowing motor events. If no swallowing events occur, this discharge activity stops, but, when swallowing is initiated, this activity amplifies, resulting in a swallowing discharge pattern (Jean, 2001). This firing pattern suggests that this group of neurones are responsible for the programming and triggering of motor events during swallowing. By contrast, the VSG comprises switching interneurones and acts as a relay unit in which neurones are activated by DSG neurones, and then send signals to motor neurones in motor nuclei that innervate the muscles in the oropharynx and oesophagus (Jean, 2001). These motor neurones are located within the trigeminal (V), facial (VII) and hypoglossal (XII) motor nuclei, the nucleus ambiguus, and the dorsal motor nucleus of the vagus (Jean, 2001).

Early studies in animals have demonstrated that swallowing can be triggered through electrical stimulation of the peripheral superior laryngeal nerve and cerebral cortex (Doty, 1951; Jean, 2001). When comparing the firing patterns of neurones in the CPG following peripheral nerve stimulation and cortical stimulation, the roles of peripheral afferents and cortical inputs on swallowing control become apparent: peripheral sensory inputs can trigger swallowing by activating neurones in the CPG, whereas the cerebral cortex, on the other hand, can initiate voluntary swallowing and control the oral preparatory (and possibly pharyngeal) stage(s) of swallowing by sending impulses to swallowing muscles through relays in the brainstem, after which swallowing is primarily mediated by the CPG in the brainstem (Jean & Car, 1979).

**Cerebral cortex and subcortical structures.** In humans, the roles of the cerebral cortex in swallowing, which is pertinent to understanding post-stroke dysphagia, have traditionally been investigated through electrical stimulation of the exposed brain during open-skull surgery and clinical lesion studies (Cheng, Takahashi, et al., 2022). Penfield & Boldrey (1937) performed the first cortical mapping study using electric stimulation of various brain regions in 120 patients who underwent neurosurgery. They found that stimulation of the anterolateral regions of primary motor cortex can elicit rhythmic swallowing, suggesting the role of cortical input in swallowing, a finding that aligned with previous studies in animals.

Recently, the involvement of the cerebral cortex in swallowing has been studied relatively non-invasively through different neurophysiology and neuroimaging techniques; for example, transcranial magnetic stimulation (TMS) and functional magnetic resonance imaging (fMRI). These techniques enable investigation of brain activity during swallowing in awake, unanaesthetised humans in less artificial and more natural settings compared to open skull surgery. TMS is a non-invasive brain stimulation technique that can induce a current within targeted brain regions through electromagnetic induction (Barker et al., 1985). This induced current can depolarise nerve cells and trigger action potentials along the corticofugal or corticospinal pathways. When the motor cortex is stimulated, the volley of action potentials travels to the peripheral muscles corresponding to the cortical region being stimulated. Once they arrive, the signals from the muscles, termed motor evoked potentials (MEPs), can be detected by surface EMG electrodes. Different properties of MEPs reflect different aspects and integrity of neural pathways, making elicitation of MEPs a powerful technique to examine neurophysiological processes.

A recent review found that multiple cortical and subcortical structures are activated during swallowing (Cheng, Takahashi, et al., 2022). These include the primary motor cortex, primary somatosensory cortex, insula, cingulate cortex, supplementary motor area, premotor cortex, auditory cortex, inferior frontal gyrus, parietooccipital and prefrontal cortex, operculum, putamen, thalamus, global pallidus, internal capsule, corpus callosum, basal ganglia, caudate, pons and midbrain, inferior parietal lobule, and the cerebellum (Cheng, Takahashi, et al., 2022). Among these regions, the primary motor cortex and primary somatosensory cortex are the two cortical regions that have been consistently identified across studies with different medical imaging techniques (Cheng, Takahashi, et al., 2022).

The primary motor cortex is responsible for the initiation and execution of swallowing. Evidence of the involvement of the primary motor cortex in swallowing came from direct electrical stimulation of the motor cortex (Penfield & Boldrey, 1937), TMS cortical mapping studies (Aziz et al., 1995; Hamdy et al., 1996) and fMRI studies (Hamdy et al., 1999; Malandraki et al., 2009; Mosier & Bereznaya, 2001). The cortical representation of the three main muscular structures involved in swallowing: the mylohyoid, and pharynx and oesophagus, was first established by Hamdy et al. (1996) using TMS. These structures are somatotopically represented in the premotor and motor cortex of both hemispheres (Fig. 1). Although they are represented bilaterally, the size of representation is different between hemispheres such that a dominant hemisphere for swallowing exists for each individual, and such dominance is independent of handedness (Hamdy et al., 1996). Moreover, an artificially induced lesion in the primary motor cortex can lead to temporary disruption in the timing and accuracy of swallow (Mistry et al., 2007; Verin et al., 2012).

The primary somatosensory cortex is thought to process afferent information during swallowing. Sensory input during swallowing is important because it provides feedback to the system to adjust muscular movements according to the properties and status of the bolus (Miller, 1982). The somatosensory and primary motor cortices are often observed to have been simultaneously activated during sensory stimulation and swallowing in fMRI studies (Cheng, Takahashi, et al., 2022). Increased activation of the primary somatosensory cortex has been reported during mechanical air-pulse stimulation (Lowell et al., 2008; Miyaji et al., 2014) or water retention (Zald & Pardo, 2000), electrical stimulation of pharyngeal mucosa (Gow et al., 2004), taste stimulation (Faurion et al., 1998) and distension of the lower oesophagus (Aziz et al., 1997).

Other subcortical structures are also involved in the neural control of swallowing, although their roles are less clear compared to those of the primary sensorimotor cortex. The insula and frontal operculum are

considered to be the primary taste centre that processes taste information and mechanical sensory information from vagal and spinal afferents (Rolls & Baylis, 1994). Studies have also suggested that the insula is a primary integrative area for volitional swallowing and may be responsible for the initiation of swallowing (Watanabe et al., 2004). The anterior cingulate cortex may be involved in the processing of visceral pain, higher cognitive level processing and attention to swallowing (Cheng, Takahashi, et al., 2022). Furthermore, the basal ganglia are part of a functional loop within the swallowing network (Lowell et al., 2012). Given their roles in the selection of appropriate motor programs, they may be responsible for regulating various motor functions during swallowing (Cheng, Takahashi, et al., 2022; Groenewegen, 2003).

**Cerebellum.** The cerebellum is a motor modulatory organ was understood to fine tune the muscle contractions that occur during swallowing (Sasegbon & Hamdy, 2021) in a similar manner to its fine tuning of limb movements (Ebner, 1998). The importance of the cerebellum in swallowing has been supported by evidence from animal studies and neurophysiological and neuroimaging studies in humans (Cheng et al., 2022; Sasegbon & Hamdy, 2021). An early TMS study showed that stimulation of the cerebellum can elicit pharyngeal MEPs and facilitate cortically induced pharyngeal MEPs (Jayasekeran et al., 2011), providing the first evidence of cerebellar–cortical neural pathways modulation in the human swallowing system. Further studies have found that the excitability of the pharyngeal motor cortex can be modulated by stimulation of the cerebellum using 10 Hz repetitive TMS (Sasegbon et al., 2021; Sasegbon et al., 2019; Sasegbon, Smith, et al., 2020; Vasant et al., 2015). These findings

suggest that the cerebellum helps modulate the central neural control of swallowing.

Evidence from neuroimaging and neurophysiological studies have provided valuable insights into the involvement of the cerebral cortex, subcortical structures and the cerebellum in the neural control of swallowing, challenging the conventional belief that the brainstem is the sole structure responsible for swallowing. Nonetheless, there remain unanswered questions regarding the functional organisation of these structures within the swallowing system and their specific roles, as well as the potential hemispheric specialisation for swallowing (Cheng, Takahashi, et al., 2022).

Taken together, the physiology of the normal swallowing process is highly complex, involving dynamic interactions between afferent and efferent pathways within the nervous system. The cerebral cortex, subcortical structures, brainstem and cerebellum play vital roles in the control of swallowing. It is therefore not surprising that damage to these structures following neurological injury (e.g. strokes) can disrupt the normal swallowing process, leading to dysphagia. The following sections will focus on the impact of strokes on swallowing, the mechanisms of recovery and the potential management options for PSD.

## The pathophysiology of PSD

Strokes, defined as syndromes of focal or global cerebral disruption originating from an underlying vascular pathological process (Sacco et al., 2013), are a commonly studied cause of neurogenic dysphagia worldwide (Cheng et al., 2021). Despite this, epidemiological studies of the incidence of PSD in the acute post stroke period are variable, with rates ranging from 3% (Kuptniratsaikul

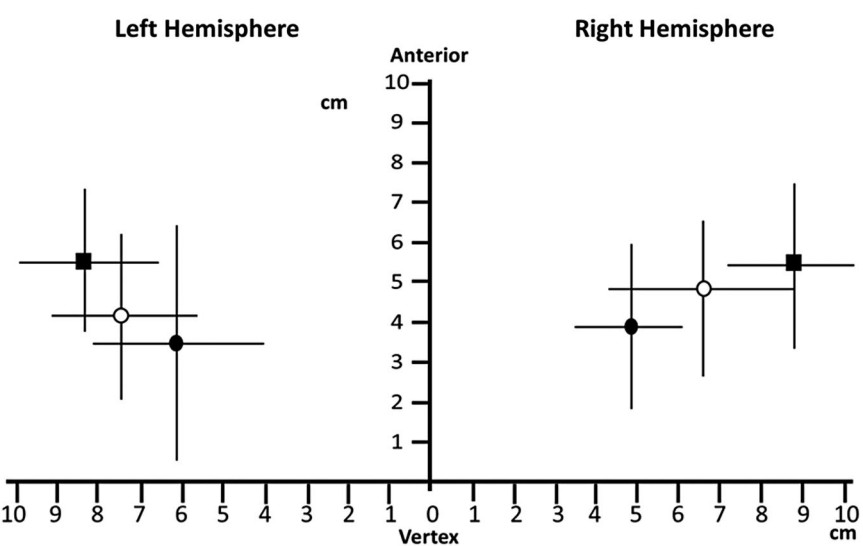

**Figure 1. Location of the representation of mylohyoid muscle, pharynx and oesophagus on the cortex**
A plot graph showing the location of the representation of the mylohyoid muscle (solid square), pharynx (empty circle) and oesophagus (solid circle) on the cortex (hotspot identified by transcranial magnetic stimulation). The lines projecting from the hotspot symbols represent the standard deviations of the distance from the vertex of the cortical representation of these muscles. The cortical representations of these muscles are arranged somatotopically in the premotor and motor cortex of both hemispheres. Reproduced with permission from Hamdy et al. (1996).

et al., 2013) to over 80% (Martino et al., 2005; Meng et al., 2000). It is now known that the occurrence of PSD is dependent on several factors including the type of stroke, lesion size and location (Jang et al., 2017; Khedr et al., 2021; Qin et al., 2023). When present, PSD has been shown to lead to increased rates of aspiration pneumonia (Sreedharan et al., 2022) and increased hospital length of stay (Arnold et al., 2016). Furthermore, the presence of dysphagia has been shown to impact clinical decisions such as discharge destination (Vasan et al., 2023) and the need for artificial feeding (Arnold et al., 2016).

Despite the high incidence of PSD in the acute post stroke period, substantial numbers of individuals recover their ability to swallow in weeks that follow, such that, at the point of discharge from hospital, the prevalence of PSD has more than halved (Sreedharan et al., 2022). Few epidemiological studies have investigated the prevalence of PSD over the long term, with the some showing a reduction in PSD prevalence, from the acute stroke phase, of over 80% (Mann et al., 1999). This natural process of dysphagia recovery is considered to be a result of neuroplastic compensatory mechanisms that were first described in humans in 1998 by Hamdy et al. (1998).

## Factors associated with dysphagia

Multiple studies have highlighted that far from being random, there are several pathophysiological factors which are associated with the occurrence of PSD and its recovery (Yang & Pan, 2022).

As mentioned earlier, the swallowing neurological pathway is complex, involving control centres at the level of the brainstem, as well as higher centres within the cerebral cortices and cerebellum (Sasegbon & Hamdy, 2017, 2021). Dysphagia can occur following damage to any of these swallowing centres (Huang et al., 2023). Few studies have been performed comparing the statistical likelihood of developing dysphagia between swallowing centres within the CNS. Some of these include a 1999 study by Daniels & Foundas (1999), which showed lesions to the primary sensory cortices had a lower probability of causing dysphagia than lesions to the primary motor cortices. Another subsequent study showed that lesions to subcortical areas had a higher probability of causing dysphagia than lesions to the cortical areas (Hess et al., 2021). Because of the lack of knowledge relating to the dynamic interactions between cortical and subcortical structures during swallowing, it is not currently known whether the observed increase in the probability of dysphagia occurring following subcortical lesions is the result of damage to subcortical areas themselves or disruptions to the connections between cortical and subcortical structures. Supplementing this information, as discussed earlier, multiple studies in healthy individuals have been performed which, when considered as a whole, have shown certain swallowing centres are more commonly identified than others. Following on from this, it can perhaps be inferred that damage to these most commonly identified brain areas most probably results in PSD. The brainstem, which is known to contain the swallowing CPG, has been shown to be activated during swallowing in less than 10% of imaging studies (Cheng, Takahashi, et al., 2022). This is rather unusual because of the extensive neurophysiological evidence in support of the central role of the brainstem in the co-ordination of the swallowing process (Jean, 2001), as well as the observation that a high proportion of brainstem strokes result in dysphagia (Meng et al., 2000). A possible explanation for this discrepancy is the comparative difficulty of measuring changes in activity within the brainstem (Beissner, 2015). Swallowing centres that have been frequently shown in the literature to cause dysphagia when damaged include the: supplementary and primary motor cortices (Galovic et al., 2016), thalamus (Hess et al., 2021), frontal lobe (Moon et al., 2018), insula (Di Stefano et al., 2021), basal ganglia (Galovic et al., 2016) and brainstem (Meng et al., 2000) (Table 2).

However, it must be noted that our understanding of the effects of cerebellar damage on dysphagia is not as clear as for other swallowing centres. Up until recently, it was understood that isolated damage to the cerebellum without brainstem involvement did not typically cause dysphagia (Flowers et al., 2011; Sasegbon & Hamdy, 2021). This view was supported by a meta-analysis conducted by Flowers et al. (2011), as well as subsequent imaging studies (Dehaghani et al., 2016; Fernández-Pombo et al., 2019). However, more recently, evidence has emerged opposing this view. Studies by Hajipour et al. (2023) and Huang et al. (2023) found that isolated cerebellar lesions can cause dysphagia in 11% and 53% of patients respectively. In particular, older patients (Hajipour et al., 2023; Huang et al., 2023) and those with multiple cerebellar lesions (Huang et al., 2023) were found to have a significantly greater probability of suffering from dysphagia. Potential explanations tying together these contrasting strands of evidence are presbyphagia and the multilimbed cerebellar motor homunculus (Sasegbon & Hamdy, 2021). Presbyphagia, defined as an age-related decline in swallowing function (Muhle et al., 2015), predisposes individuals to dysphagia as a result of reducing their functional reserve. In essence, only a small push, such as by an isolated cerebellar lesion, may be required to tip a presbyphagic patient into frank dysphagia. The cerebellar motor homunculus is unlike the more typically known cortical motor homunculus. Muscles, including oral and pharyngeal muscles, appear in multiple places on both hemispheres and the vermis (Sasegbon & Hamdy, 2021). Were it to be constructed, it would look akin to a sea star or spider (Sasegbon & Hamdy, 2021). Therefore, it

**Table 2. Lesion location and aftereffects**

| Location of lesion | Studies | Imaging modality | Number of participants | Type of stroke | Effects |
|---|---|---|---|---|---|
| Primary motor cortex | Suntrup et al. (2015) | MRI/CT | 200 | Ischaemic + | Aspiration |
| | Suntrup et al. (2017) | MRI/CT | 200 | Haemorrhagic | Prolonged oral |
| | Daniels & Foundas (1999) | MRI/CT | 54 | Ischaemic + | transit time (OTT) |
| | Jang et al. (2017) | MRI | 82 | Haemorrhagic | Oral residue |
| | | | | Ischaemic | Pharyngeal residue |
| | | | | Ischaemic + | Delayed SR |
| | | | | Haemorrhagic | |
| Supplementary motor area | Daniels & Foundas (1999) | MRI/CT | 54 | Ischaemic | Aspiration |
| Primary somatosensory cortex | Suntrup et al. (2015) | MRI/CT | 200 | Ischaemic + | Aspiration |
| | Suntrup et al. (2017) | MRI/CT | 200 | Haemorrhagic | Pharyngeal residue |
| | Daniels & Foundas (1999) | MRI/CT | 54 | Ischaemic + | Impaired laryngeal |
| | | | | Haemorrhagic | vestibule (LV) |
| | | | | Ischaemic | closure |
| Insula | Wilmskoetter et al. (2019) | MRI | 68 | Ischaemic | Aspiration |
| | Hess et al. (2021) | MRI/CT | 132 | Haemorrhagic | Prolonged OTT |
| | Im et al. (2018) | MRI | 21 | Ischaemic + | Prolonged |
| | Galovic et al. (2013) | MRI | 94 | Haemorrhagic | pharyngeal transit |
| | Galovic et al. (2017) | MRI | 62 | Ischaemic | time (PTT) |
| | Moon et al. (2018) | MRI | 90 | Ischaemic | Impaired laryngeal |
| | Kim et al. (2022) | MRI | 100 | Undefined | elevation (LE) |
| | Hu et al. (2022) | MRI | 126 | Ischaemic + | Impaired LV closure |
| | | | | Haemorrhagic | |
| | | | | Ischaemic | |
| Thalamus | Im et al. (2018) | MRI | 21 | Ischaemic + | Aspiration |
| | | | | Haemorrhagic | |
| Basal ganglia | Im et al. (2018) | MRI | 21 | Ischaemic + | Aspiration |
| | Galovic et al. (2013) | MRI | 94 | Haemorrhagic | Prolonged OTT |
| | Daniels & Foundas (1999) | MRI/CT | 54 | Ischaemic | Prolonged PTT |
| | Jang et al. (2017) | MRI | 82 | Ischaemic | Delayed swallowing |
| | Moon et al. (2018) | MRI | 90 | Ischaemic + | reflex (SR) |
| | Kim et al. (2022) | MRI | 100 | haemorrhagic | |
| | Nakamori et al. (2021) | MRI | 342 | Undefined | |
| | | | | Ischaemic + | |
| | | | | Haemorrhagic | |
| | | | | Ischaemic + | |
| | | | | Haemorrhagic | |
| Operculum | Suntrup et al. (2015) | MRI/CT | 200 | Ischaemic + | Aspiration |
| | Suntrup et al. (2017) | MRI/CT | 200 | Haemorrhagic | Oral residue |
| | Galovic et al. (2013) | MRI | 94 | Ischaemic + | Prolonged dysphagia |
| | Galovic et al. (2016) | MRI | 119 | Haemorrhagic | |
| | Galovic et al. (2017) | MRI | 62 | Ischaemic | |
| | | | | Ischaemic | |
| | | | | Ischaemic | |
| Brainstem | Daniels & Foundas (1999) | MRI/CT | 54 | Ischaemic | Aspiration |
| | Jeon et al. (2014) | MRI/CT | 178 | Ischaemic + | Oral impairment |
| | Daniels et al. (2017) | MRI | 80 | Haemorrhagic | Pharyngeal |
| | Hu et al. (2022) | MRI | 126 | Ischaemic | impairment |
| | | | | Ischaemic | |
| Cerebellum | Huang et al. (2023) | MRI/CT | 1651 | Ischaemic + | Aspiration |
| | | | | Haemorrhagic | |

Abbreviations: LE, laryngeal elevation; LV, laryngeal vestibule; OTT, oral transit time; PTT, pharyngeal transit time; SR, swallowing reflex.

may be that the cerebellum, in isolation, needs to be more heavily damaged for decompensation to occur. This may explain the observation that multiple cerebellar lesions have a greater probability of causing dysphagia. Therefore, although it may be the case that a younger patient without presbyphagia does not become dysphagic after an isolated cerebellar lesion, if presbyphagia is present or multiple lesions occur, then dysphagia may manifest.

Haemorrhagic strokes have been shown by multiple studies to have a greater probability of causing dysphagia than ischaemic strokes (Khedr et al., 2021; Vasan et al., 2023), with a recent meta-analysis showing patients with ischaemic strokes have half the probability of developing PSD compared to patients with haemorrhagic strokes (Banda et al., 2022). The precise reason for this is currently unknown, but it is currently assumed that this observation is related to the increased size of haemorrhagic compared to ischaemic lesions (Jørgensen et al., 1995). The larger the lesion, the greater its deleterious effect hence the higher National Institutes of Health Stroke Scale (NIHSS) scores in patients with haemorrhagic compared to ischaemic strokes, as well as their longer hospital length of stays and increased mortality (Vasan et al., 2023).

Studies have also shown that the risk of PSD increases with the severity of the stroke as measured using the total NIHSS score (Khedr et al., 2021). In addition, studies have shown that increased stoke size (Khedr et al., 2021) and, in the case of cortical strokes, the presence of bilateral damage (D'Netto et al., 2023) increases the risk of PSD.

Other factors which have been shown to be associated with PSD include increased age, hypertension and diabetes (Yang & Pan, 2022). The data regarding gender is mixed with some studies showing an increased likelihood of PSD in men compared to women (Henke et al., 2017), whereas others show the opposite (Arnold et al., 2016). A recent meta-analysis on the topic did not show a significant association (Yang & Pan, 2022).

In summary, although more is now known of the various factors that increase the likelihood of developing PSD, there remain gaps in our understanding. Examples of these gaps include contrasting evidence with respect to gender as a risk factor for PSD. Another example is the discrepancy between our established neurophysiological understanding of the key co-ordinating role of the brainstem in the process of swallowing and its infrequent identification in swallowing functional imaging studies. More studies need to be performed to address these issues.

### Recovery of PSD

Post-lesional neuroplastic changes in damaged and undamaged areas of the brain (Hamdy et al., 1998) are considered to be the underlying reason for the recovery of swallowing function. Functional imaging studies have shown ipsilesional brain changes occurring in patients with PSD (Li et al., 2009; Malandraki et al., 2011) and limb dysfunction (Nelles et al., 1999). However, specifically regarding recovery in patients with PSD, the contralesional sensorimotor cortex has been observed to exhibit a greater increase in activity than was observed over the affected hemisphere (Hamdy et al., 1998). This observation was assumed to be compensation for the damage caused to its hemispheric partner. This contra-lesional compensatory process was first shown by Hamdy et al. (1998) and has also been observed to occur in subsequent PSD functional imaging studies (Almeida et al., 2017; Li et al., 2009; Mihai et al., 2016). In the study by Hamdy et al. (1998), 28 patients with unilateral cortical strokes were followed up over 3 months. PSD was observed in 71% of patients in the immediate post stroke period, which reduced to 41% at 3 months (Hamdy et al., 1998). Those patients with PSD had smaller TMS determined pharyngeal representations over their affected hemispheres, indicating neuronal damage (Hamdy et al., 1998). In those patients who recovered, an increase in pharyngeal activity was predominantly seen over their undamaged hemispheres (Hamdy et al., 1998). By contrast, those patients in whom PSD persisted did not exhibit observable compensatory changes (Hamdy et al., 1998).

On a cellular level, several cytokine controlled mechanisms in the brain work towards compensating for and, in some instances, repairing damage (Regenhardt et al., 2020). The key elements of this process, which ultimately lead to changes ipsilaterally in the areas of the brain surrounding the lesion (Overman et al., 2012), as well as distantly (Brown et al., 2009) and contralaterally (Dodd et al., 2017), can be summarised thusly. After damage occurs, studies in mouse models have observed an initial reduction in dendritic spines in the area of the brain surrounding the lesion followed, as time progresses, by increased production over and above their previous baseline (Brown et al., 2008). Additionally, there is growth of new axons (Li et al., 2015). This process has been shown to occur around the lesion, as well as in more distant functionally associated regions in the brains of mice (Brown et al., 2009; Li et al., 2015). In humans, this process is also thought to occur in the peri-lesional region because of the upregulation of the growth factor producing gene GAP43 (Overman et al., 2012). The role that neurogenesis plays in repair and recovery in the human brain is currently unclear (Regenhardt et al., 2020). Arteriogenesis and angiogenesis have also been observed to occur ipsi- and contralesionally in mouse models (Liu et al., 2014) and humans in the post-lesional brain (Krupinski et al., 1994). In humans, a small study of 10 patients has shown that post-lesional angiogenesis is correlated with increased survival (Krupinski et al., 1994).

In parallel to the cellular changes that occur, there are fluctuant changes to the overall excitability of the ipsi- and contralesional brain (Dodd et al., 2017). Following damage, studies have shown that there is an initial increase in neuronal excitability above baseline (Fujioka et al., 2004) followed by a period of relative inhibition (Clarkson et al., 2010). This changing excitability is considered to act as the catalyst for long-term changes in neuronal function (Carmichael, 2012), presumably through the processes of long term potentiation (LTP) and depression (LTD) (Bliss & Gardner-Medwin, 1973; Ito & Kano, 1982). LTP and LTD are known to be the driving forces behind the process of neuroplasticity and involve receptors such as NMDA and AMPA and the process of phosphorylation in the short to medium term, and, ultimately, altered gene expression and protein production (Lee et al., 2003).

Despite the evidence presented above from functional imaging and neurophysiological studies regarding ipsi- and contralesional changes in brain activity and excitability, few PSD studies have been performed observing the evolution of post-lesional brain activity (including within the brainstem and cerebellum) and swallowing changes. Studies which have been published in this area, such as the study by Hamdy et al. in (1998), have only assessed this changes in relatively infrequent snapshots. This is an area of our understanding that requires further strengthening; for example, by a functional imaging study incorporating weekly swallowing assessment in conjunction with neurophysiological or imaging studies over the course of a patient with PSD's admission to hospital.

The central aim of neuromodulatory techniques in the field of PSD is to induce compensation and neuroplastic recovery in those patients who do not and would not naturally compensate. Strokes are an important yet comparatively straightforward disease process in which to employ these novel techniques because, for the most part, there is a single static insult around which compensation can be induced. By contrast, in other forms of neurogenic dysphagia such as Parkinson's disease, there is a continuous process of neuronal damage with the potential to reverse any beneficial neuroplastic changes that are therapeutically induced (Sasegbon et al., 2021).

### Current management

The current approach to managing individuals who do not recover their ability to swallow naturally involves three strategies: compensation, rehabilitation and artificial feeding. Compensation, for the most part, involves modifying the texture of foods and fluids (Steele et al., 2015). However, the evidence as to the effectiveness of this approach is weak and often contradictory despite its widespread use (Hansen et al., 2022). Rehabilitation

involves patients being taught exercises or manoeuvres to improve the efficiency and safety of their swallowing (Cheng, Sasegbon, et al., 2022) (Robbins et al., 2007). Rehabilitative interventions have been shown to lead to significant improvements in oropharyngeal muscle strength and bulk (Park et al., 2019). They have been used for decades as a means to improve quality of life in patients with PSD and currently constitute the cornerstone of neurogenic dysphagia treatment in clinical settings. Functional imaging studies have also shown increased brain activity in swallowing areas following rehabilitative exercises (Peck et al., 2010). Hence, these exercises act not only to improve oropharyngeal muscle strength, but also to modulate the neuronal control of those strengthened muscles. However, it should be noted that, despite rehabilitative exercises forming a key part of the clinical treatment armamentarium, there are relatively few studies that have aimed to investigate the impact of different rehabilitative exercises on swallowing safety. Furthermore, no studies have investigated the effect on dysphagia of rehabilitative exercises used alongside novel neuromodulatory interventions. These large gaps in our understanding will need to be addressed in future studies. Standard rehabilitative care can be supplemented with biofeedback, with some studies showing that this improves muscle contractility (Benfield et al., 2019).

Should the above approaches fail and a patient's swallow remain unsafe, artificial feeding can be considered. This mainly consists of nasogastric feeding and percutaneous gastrostomy or jejunostomy insertion (Sutcliffe et al., 2020). However, it must be noted that, although artificial feeding improves nutrition and hydration, prepyloric feeding does not eliminate the risk of aspiration (Sutcliffe et al., 2020). None of these more traditional management approaches aim to reverse the neuronal damage underpinning PSD and restore swallowing. However, the restoration of normal swallowing functionality is the central aim of all neuromodulatory interventions.

### Neuromodulation

There are various neuromodulatory techniques which can be used to adjust neuronal activity within targeted brain regions by harnessing the processes of LTP and LTD. These techniques can be broadly grouped into peripheral techniques such as pharyngeal electrical stimulation (PES) (Michou et al., 2014) and neuromuscular electrical stimulation (Lim et al., 2014), and centrally acting techniques such as repetitive transcranial magnetic stimulation (rTMS) (Michou et al., 2014), transcranial direct current stimulation (TDCS) (Zhang et al., 2021), transcranial alternating current stimulation (Zhang et al., 2021) and transcranial random noise stimulation (Zhang et al., 2021). The techniques that have the greatest

evidence base in PSD are PES, the first neuromodulatory technique to be developed in the swallowing motor system, rTMS, the technique with the greatest amount of evidence in support of its use, and TDCS (Cheng et al., 2021).

PES involves inserting a catheter containing two ring electrodes either trans-nasally or trans-orally and positioning it within the pharynx (Fraser et al., 2003). Once it is adequately positioned, it is connected to a signal generator and activated. A small electrical current at a frequency of 5 Hz is then passed through the catheter to the surrounding mucosa (Fraser et al., 2003). The mucosal effects of PES, as described above, lead to increased sensory and subsequent indirect motor cortical modulation of swallowing neurological pathways (Fraser et al., 2003). TDCS is a centrally acting technique that modulates brain activity by impacting the resting membrane potential of targeted neurones (Zhang et al., 2021). The equipment comprises two electrodes and a signal generator. The anode is placed atop a sponge soaked in saline and positioned over the area of the brain that requires stimulation (Sasegbon, Cheng, et al., 2020). The cathode is prepared in a similar manner and is typically placed over the contralateral brow (Sasegbon, Cheng, et al., 2020). When the signal generator is switched on, direct current flows between the electrodes. This causes excitation if the anode is placed over the motor cortices, or suppression if the cathode is similarly positioned (Sasegbon, Cheng, et al., 2020). RTMS is a centrally acting technique that aims to alter neuronal activity using pulses of electromagnetic energy. It involves the use of a signal generator attached to an electromagnetic coil, with the most commonly used coil in dysphagia research being a figure of eight coil (Vasant et al., 2015). When high frequency rTMS is administered over swallowing centres, neuronal excitation occurs, whereas low frequency rTMS causes neuronal suppression. In the swallowing motor system, 5 Hz applied over the primary motor cortex causes excitation (Michou et al., 2014), whereas 10 Hz has a similar excitatory effect when applied over the cerebellum (Sasegbon et al., 2019).

Given the mechanism of neuroplastic recovery in PSD described above, there are six broad treatment approaches that have been employed by researchers to induce compensatory brain changes. The first is to apply neuroexcitatory techniques such as high frequency rTMS or anodal TDCS to the unaffected cortical hemisphere to provoke compensation in a manner that mirrors the natural process of compensation described above (Michou et al., 2014). The second is the use of the aforementioned neuroexcitatory techniques to target the swallowing centre over the damaged cortical hemisphere in an attempt to directly reverse the post stroke damage and restore, as much as is possible, normal bi-hemispheric neurophysiology (Yang et al., 2012). The third is to use neuro-excitatory techniques bilaterally over the affected and unaffected cortical hemispheres to both induce contra-lesional compensation and provoke ipsilesional recovery (Momosaki et al., 2014). The fourth is to use neuro-suppressive techniques such as low frequency rTMS over the unaffected hemisphere to theoretically prevent it from affecting the recovery of its damaged partner (Verin & Leroi, 2009). The fifth approach is a newer approach that aims to use excitatory techniques to target the cerebellum and, by so doing, not only modulate cerebellar neuronal activity, but also indirectly modulate cortical and presumably brainstem neuronal activity (Vasant et al., 2019). The sixth approach is peripheral and uses techniques such as PES to increase sensory inflow into the swallowing sensorimotor system (Michou et al., 2014). This approach aims to modulate activity within the sensory cortices, which in turn will modulate activity and induce neuroplastic changes within the supplemental motor and primary motor cortices.

Recent meta-analyses show that there is some evidence that these techniques are effective over the short and medium term at improving PSD (Cheng et al., 2021; Zhu & Gu, 2022) with contralesional and bilateral stimulation appearing to be more effective than ipsilesional stimulation in PSD (Cheng et al., 2021). However, it must be noted that because of the comparatively limited number of studies in the field, the studies incorporated into the metanalyses were heterogeneous with respect to their treatment protocols and outcome measures. Furthermore, recent evidence from cerebellar rTMS studies has indicated that bilateral stimulation is superior to ipsi- or contralesional stimulation with respect to reversing the effects of PSD (Dai et al., 2023). However, regardless of the increasing wealth of data, there remain significant gaps in our knowledge in terms of fully implementing these newer technologies into mainstream clinical management in the neurorehabilitation of dysphagia after stroke. Going forward, more studies will need to be performed employing the neuromodulatory approaches and protocols with the greatest amount of evidence in support of their effectiveness in patients with acute, subacute and chronic PSD. This will reduce and may eliminate the current problem in the field, which was alluded to above in that relatively few studies exist, many of which target different oropharyngeal muscles and hemispheric areas and use different stimulation energies and durations of treatment. Should this be done, a more comprehensive understanding of the true effect of neuromodulation on PSD could be achieved.

## Summary

The neuronal control and muscular contractility required for swallowing is highly complex and in its complexity is liable to be damaged by a variety of insults.

Neuroplasticity is the means by which natural recovery occurs should there be damage to swallowing centres within the CNS. Unfortunately, not all people are able to compensate naturally and are subject to alterations to the consistency of their food and fluids and rehabilitative exercises with the aim of improving their swallowing safety. With the advent of neuromodulatory techniques, it is now possible, by harnessing the process of neuro-plasticity, to induce compensatory changes in those who would not naturally do so. Notwithstanding, larger clinical trials of these new and emerging techniques are still required before full implementation.

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

## Additional information

### Competing interests

SH is the chief scientific officer and is on the board of Phagenesis Ltd, a company involved in neuromodulatory dysphagia treatment. The other authors declare that they have no competing interests.

### Author contributions

A.S. and I.C. drafted the manuscript together and are joint first authors. S.H. supervised the drafting of the manuscript and contributed to its structure. All authors made contributions to the manuscript and jointly submitted it for publication

### Funding

No funding was received.

### Keywords

dysphagia, neuromodulation, neuroplasticity, PES, rTMS, strokes, swallowing, TDCS

## Supporting information

Additional supporting information can be found online in the Supporting Information section at the end of the HTML view of the article. Supporting information files available:

**Peer Review History**

