## [Peer Review History · The Journal of Physiology]

The neurorehabilitation of post-stroke dysphagia - physiology and pathophysiology

Ayodele Sasegbon, Ivy Cheng, and Shaheen Hamdy

DOI: 10.1113/JP285564

Corresponding author(s): Shaheen Hamdy (shaheen.hamdy@manchester.ac.uk)

Review Timeline:

Submission Date:	02-Dec-2023
Editorial Decision:	05-Feb-2024
Revision Received:	23-Feb-2024
Accepted:	29-Feb-2024

Senior Editor: Laura Bennet

Reviewing Editor: Richard Carson

Transaction Report:

Dear Professor Hamdy,

Re: JP-TR-2023-285564 "The neurorehabilitation of post-stroke dysphagia - physiology and pathophysiology" by Ayodele Sasegbon, Ivy Cheng, and Shaheen Hamdy

Thank you for submitting your manuscript to The Journal of Physiology. It has been assessed by a Reviewing Editor and by 1 expert referee and we are pleased to tell you that it is acceptable for publication following satisfactory revision.

ABSTRACT FIGURES: Authors may use The Journal's premium BioRender account to create/redraw their Abstract Figures (and any other suitable schematic figure). Information on how to access this account is here: <https://physoc.onlinelibrary.wiley.com/journal/14697793/biorender-access>.

REVISION CHECKLIST: Upload a full Response to Referees file. To create your 'Response to Referees' copy all the reports, including any comments from the Senior and Reviewing Editors, into a Microsoft Word, or similar, file and respond to each point, using font or background colour to distinguish comments and responses and upload as the required file type.

We look forward to receiving your revised submission.

Yours sincerely,

EDITOR COMMENTS

Reviewing Editor:

The paper is comprehensive, and analyses not only the physiological processes relevant to the pathology, but also their clinical significance and implications with respect to strategies of rehabilitation. As the referee emphasises, the potential impact of the piece may be enhanced further by attending to several points for which a different emphasis may be required, and to instances in which there is a need for circumspection - given the current evidential base is limited.

It is therefore recommended that the authors incorporate the suggestions made by the referee - all of which have been made with positive intent.

Please also see 'Required Items' below.

REFEREE COMMENTS

Referee #1:

Thank you for the opportunity to review this important and well-written manuscript! Stroke is the leading cause of dysphagia and while we know a lot, there is a lot left to learn. To this point, it would be helpful if the authors include information related to the strength of the findings included. Some of this information is presented as fact, but the result of a single, rather small sample size, study. Furthermore, it would be helpful to include a section on what we do not yet have definitive evidence to support. This could be added to the end of each subsection (neurophysiology of swallowing, factors associated with dysphagia, recovery of post stroke dysphagia, current management, & neuromodulation). Please see line-item comments below.

Abstract:

Line 43: Suggestion - PSD is not only the most commonly studied form of dysphagia but is also the most prevalent form as stroke is the leading cause of dysphagia

Line 106: I believe this is referring to stage III (not II).

Line 109: As the original sources of this reference rely on data from a small number of dogs and monkeys, it is suggested that the authors generally state that suprahyoid muscle contraction begins the pharyngeal stage. Even the provided reference states that "at the same time or after a very short delay of 30-40ms, a contraction also begins in ..." other suprahyoid muscles.

Line 112: I do not believe that "partial leakage" is the most accurate representation of this process. It is most typical that it is stated as "pharyngeal swallow is triggered by bolus pressure to sensory receptors in the posterior oropharynx". Please consider changing the wording to not include the term "leakage".

Line 113: Please include language to indicate that this is describing a normal swallow as these terms used, "invariable sequence" (also line 134) and "all or none", are untrue for a disordered swallow where we often see aspects of pharyngeal swallow being impaired while other aspects are typically functioning.

Lines 176-177: The data that this statement is based on "males exhibit more significant deterioration of sensation than females" comes from a perceptual study that included only 10 older men and 10 older women. Within that study, the mean of exponents for the oral perceptions of fluid viscosity by gender for the oldest group were 0.24447 (0.0414) for men and 0.2938 (0.0269) for women. Given this small sample size and the unknown actual clinical importance of this, rather small, magnitude of difference between men and women, I suggest that this statement be hedged, for example, including a phrase stating that "there is a small amount of evidence to suggest, ..."

Lines 177-179: The data that this statement is based on "duration of apnea decreases in males but increases in females" comes from a study that included only 10 men and 10 women in each age group. Furthermore, the duration of the apnea period increased for both males and females with age when swallowing boluses of various volumes (10 mL, 15 mL, 20 mL, and 25 mL) only for the saliva swallows did the apnea duration differ between males and females. Given the results that this statement is based on and the unknown actual clinical importance of this, rather small, and specific, difference between men and women, I suggest that this statement be hedged, for example, including a phrase stating that "there is a small amount of evidence to suggest, ... in saliva swallows"

Line 259: This sentence refers to the pharynx and oesophagus as muscles. Please refine the sentence to either read "and muscles of the pharynx and oesophagus" or provide specific muscles.

Line 336: In reference to whether subcortical lesions are more likely than cortical to cause swallowing impairment, there is a lacking statement / reference to the importance of, and relatively unmeasured / unresearched, underlying neural network between the cortical and subcortical areas such as it is unknown how much the swallowing problem is due to a lesion in the subcortical area versus the disconnection the lesion caused between a cortical and subcortical area. It would perhaps be helpful to add verbiage related to this.

Line 375: It has been at least postulated that the reason for haemorrhagic strokes resulting in dysphagia more frequently than ischaemic strokes is that they result in larger lesions (hence greater impairment)

In reference to Line 389 "neuroplastic compensation is the process that underlies the recovery of swallowing", this statement does not account for recovery / repair in the area of the lesion (which is supported in the following paragraph referencing knowledge from animal model lesion and recovery studies). Furthermore, it is a rather strong statement being that it is based off of a study with only 11 patients with post-stroke dysphagia at the three month recovery point. It is suggested to rephrase to better represent the state of the evidence.

Line 446: This phrase "rehab only provides modest improvements to swallow safety" is supported by only one study and does not accurately characterize the impact of rehabilitation on countless patients with post-stroke dysphagia. It is erroneous, and frankly dangerous, to suggest as a blanket statement that rehab does not help. While it might not help all recover to be able to eat fully by mouth, especially those with very severe conditions, it does remarkably change the lives of others. I strongly suggest that this statement be modified to reflect that with improved knowledge to harness neuroplasticity that we could optimize rehabilitation results for all with post-stroke dysphagia and to help those with very severe conditions that are currently unable to eat by mouth. (Note: please be careful with these types of erroneous blanket statements, in the U.S. insurance providers will use this to start denying rehabilitation access to patients with post-stroke dysphagia. We do so much good to help so many patients, it would be truly horrible if that happened.)

Line 453: This statement misses that a fundamental goal of rehabilitation exercises is to modify the neurophysiology of swallowing. It is hard to imagine how exercises to improve swallowing, that are sufficiently effective, wouldn't result in neurological / cortical changes. Similarly, the goal of neurophysiological changes due to neuromodulation without resulting changes in swallowing function would not be relevant. Please nuance this statement.

Lines 508-509: Related to the statement that meta-analyses have demonstrate that all these neurostim techniques are effective at improving PSD in the short and medium term, there are two meta-analyses provided to support this. Both meta-analyses are based on the rather limited (in terms of sample size for the individual studies) data. All of the studies included in the meta-analyses use very different neurostimulation techniques and methods as well as different outcome variables. Given the state of the currently available evidence that precludes the conduct of a meta-analysis combining studies of similar methods and similar outcomes, it is suggested that this statement be nuanced to read that there is emerging evidence that neurostimulation techniques hold promise for improving PSD.

REQUIRED ITEMS

- Please include an Abstract Figure file, as well as the Figure Legend text within the main article file. The Abstract Figure is a piece of artwork designed to give readers an immediate understanding of the Review Article and should summarise the main conclusions. If possible, the image should be easily 'readable' from left to right or top to bottom. It should show the physiological relevance of the Review so readers can assess the importance and content of the article. Abstract Figures should not merely recapitulate other figures in the Review. Please try to keep the diagram as simple as possible and without superfluous information that may distract from the main conclusion of the Review. Abstract Figures must be provided by authors no later than the revised manuscript stage and should be uploaded as a separate file during online submission labelled as File Type 'Abstract Figure'. Please ensure that you include the figure legend in the main article file. All Abstract Figures will be sent to a professional illustrator for redrawing and you may be asked to approve the redrawn figure before your paper is accepted.

- The reference list must be in alphabetical order, rather than numbered, to comply with our Journal format.

- Please upload separate high quality figure files via the submission form.

- Author profile(s) must be uploaded via the submission form. Authors should submit a short biography (no more than 100 words for one author or 150 words in total for two authors) and a portrait photograph of the two leading authors on the paper. These should be uploaded and clearly labelled together in a Word document with the revised version of the manuscript. Any standard image format for the photograph is acceptable, but the resolution should be at least 300 DPI and preferably more. A group photograph of all authors is also acceptable, providing the biography for the whole group does not exceed 150 words.

- It is the authors' responsibility to obtain any necessary permissions to reproduce previously published material and to list these within the main article file. For information, please see: https://jp.msubmit.net/cgi-bin/main.plex?form_type=display_requirements#permissions.

END OF COMMENTS

Confidential Review

02-Dec-2023

Response to comments

Dear Editor,

Thank you for taking the time to go through our paper. We have carefully read through all the comments and have made the suggested changes to the revised manuscript. These can be seen below. We trust these changes now make our paper acceptable for publication in the Journal of Physiology (special edition).

REFEREE COMMENTS:

Thank you for the opportunity to review this important and well-written manuscript! Stroke is the leading cause of dysphagia and while we know a lot, there is a lot left to learn. To this point, it would be helpful if the authors include information related to the strength of the findings included. Some of this information is presented as fact, but the result of a single, rather small sample size, study. Furthermore, it would be helpful to include a section on what we do not yet have definitive evidence to support. This could be added to the end of each subsection (neurophysiology of swallowing, factors associated with dysphagia, recovery of post stroke dysphagia, current management, & neuromodulation). Please see line-item comments below.

Thank you for your comments and suggestions. A section on evidence have been added to each subsection:

Neurophysiology of swallowing: "Evidence from neuroimaging and neurophysiological studies have provided valuable insights into the involvement of the cerebral cortex, subcortical structures and the cerebellum in the neural control of swallowing, challenging the conventional belief that the brainstem is the sole structure responsible for swallowing. Nonetheless, there remain unanswered questions regarding the functional organization of these structures within the swallowing system and their specific roles, as well as the potential hemispheric specialisation for swallowing [39]."

"In summary, although more is now known of the various factors which increase the likelihood of developing PSD, there remain gaps in our understanding. Examples of these gaps include contrasting evidence with respect to gender as a risk factor for PSD. Another example is, the discrepancy between our established neurophysiological understanding of the key coordinating role of the brainstem in the process of swallowing and its infrequent identification in swallowing functional imaging studies. Due to the strength of neurophysiological evidence from earlier animal and human studies, it suggests that difficulty functionally assessing neurological activity within the brainstem is the culprit. More studies need to be performed to address these issues."

"Despite the evidence presented above from functional imaging and neurophysiological studies regarding ipsi and contralesional changes in brain activity and excitability, few PSD studies have been

performed observing the evolution of post-lesional brain (including brainstem and cerebellum) and swallowing changes over time. Studies such as the Hamdy et al study (Hamdy et al., 1998) which have attempted this do so in relatively infrequent snapshots. This is an area of our understanding that requires further strengthening for example by a functional imaging study incorporating weekly swallowing assessment in conjunction with neurophysiological or imaging studies over the course of a patient with PSD's admission to hospital."

"However, it should be noted that despite rehabilitative exercises forming a key part of the treatment armamentarium in clinical practice, there are few studies that have sought to investigate the impact of different rehabilitative exercises on swallowing safety. Furthermore, no studies have investigated the effect on dysphagia of rehabilitative exercises used alongside novel neuromodulatory interventions. These large gaps in our understanding will need to be addressed by future studies."

"Going forwards, more studies will need to be performed employing the neuromodulatory approaches and protocols with the most amount of evidence in support of their effectiveness, in patients with acute, subacute, and chronic PSD. This will reduce and may eliminate the current problem in the field which was alluded to above wherein relatively few studies exist, many of which target different oropharyngeal muscles and hemispheric areas and use different stimulation energies and durations of treatment. Should this be done, a more comprehensive understanding of the true effect of neuromodulation on PSD could be achieved."

Abstract:

Line 43: Suggestion - PSD is not only the most commonly studied form of dysphagia but is also the most prevalent form as stroke is the leading cause of dysphagia

Thank you for the suggestion. The statement has been revised.

Line 106: I believe this is referring to stage III (not II).

Stage II Transport is a specific stage described in the Process Model for swallowing solid food to highlight the dynamic process between mastication of solid food and propulsion of chewed food. The sentences have been revised as follows:

"This is followed by Food Processing stage, where food is reduced in size through mastication and mixed with saliva [5, 7], and Stage II Transport, which is equivalent to oral propulsive stage for liquids. Food Processing may occur again following Stage II Transport, depending on if there is food remaining in the oral cavity [7]."

Line 109: As the original sources of this reference rely on data from a small number of dogs and monkeys, it is suggested that the authors generally state that suprahyoid muscle contraction begins the pharyngeal stage. Even the provided reference states that "at the same time or after a very short delay of 30-40ms, a contraction also begins in ..." other suprahyoid muscles.

Thank you for the suggestion. The statement has been revised accordingly:

"The pharyngeal stage is involuntary and is marked by the contraction of suprahyoid muscles, which constitute part of the "leading complex" that is responsible for the initiation of swallowing reflex [5]."

Line 112: I do not believe that "partial leakage" is the most accurate representation of this process. It is most typical that it is stated as "pharyngeal swallow is triggered by bolus pressure to sensory receptors in the posterior oropharynx". Please consider changing the wording to not include the term "leakage".

Thank you for the suggestion. The statement has been revised:

"The swallowing reflex is triggered by the pressure from the bolus as it reaches the receptive regions innervated by the superior laryngeal nerve during oral propulsive stage [10]."

Line 113: Please include language to indicate that this is describing a normal swallow as these terms used, "invariable sequence" (also line 134) and "all or none", are untrue for a disordered swallow where we often see aspects of pharyngeal swallow being impaired while other aspects are typically functioning.

Thank you for the suggestion. The statements have been revised:

"Once triggered, in a normal swallow, an invariable sequence of motor events occurs in an "all-or-none" manner, irrespective of the bolus characteristics, suggesting the presence of central pattern generator for swallowing [11]."

"Although the sequence of motor events during pharyngeal swallow is invariable in a normal swallow, the timing and duration of these events may be altered by descending signals from the motor cortex or the brainstem, which may be driven by sensory inputs from the oropharyngeal muscles with respect to the bolus [10]."

Lines 176-177: The data that this statement is based on "males exhibit more significant deterioration of sensation than females" comes from a perceptual study that included only 10 older men and 10 older women. Within that study, the mean of exponents for the oral perceptions of fluid viscosity by gender for the oldest group were 0.24447 (0.0414) for men and 0.2938 (0.0269) for women. Given this small sample size and the unknown actual clinical importance of this, rather small, magnitude of difference between men and women, I suggest that this statement be hedged, for example, including

a phrase stating that "there is a small amount of evidence to suggest, ..."

Thank you for the suggestion. The statements have been revised:

"Finally, there is a small amount of evidence which comes from mainly studies with small sample sizes that suggest that there may be interaction effects between age and gender on swallowing physiology. Some studies reported that males exhibit a more significant deterioration in sensitivity of fluid viscosity with increasing age than females [33], and that the duration of swallowing cessation (the holding of breath at the point of swallowing) during saliva swallows decreases with increasing age in males but increases in females [34]."

Lines 177-179: The data that this statement is based on "duration of apnea decreases in males but increases in females" comes from a study that included only 10 men and 10 women in each age group. Furthermore, the duration of the apnea period increased for both males and females with age when swallowing boluses of various volumes (10 mL, 15 mL, 20 mL, and 25 mL) only for the saliva swallows did the apnea duration differ between males and females. Given the results that this statement is based on and the unknown actual clinical importance of this, rather small, and specific, difference between men and women, I suggest that this statement be hedged, for example, including a phrase stating that "there is a small amount of evidence to suggest, ... in saliva swallows"

Thank you for the suggestion. The statements have been revised as stated in the response to the comment above.

Line 259: This sentence refers to the pharynx and oesophagus as muscles. Please refine the sentence to either read "and muscles of the pharynx and oesophagus" or provide specific muscles.

Thank you for the suggestions. The statement has been revised:

"The cortical representation of three main muscles involved in swallowing: and the mylohyoid, and muscles of the pharynx and oesophagus, was first established by Hamdy and colleagues using TMS [43]."

Line 336: In reference to whether subcortical lesions are more likely than cortical to cause swallowing impairment, there is a lacking statement / reference to the importance of, and relatively unmeasured / unresearched, underlying neural network between the cortical and subcortical areas such as it is unknown how much the swallowing problem is due to a lesion in the subcortical area versus the disconnection the lesion caused between a cortical and subcortical area. It would perhaps be helpful to add verbiage related to this.

Thank you. The text has been changed to:

“Due to the lack of knowledge relating to the dynamic interactions between cortical and subcortical structures during swallowing, it is not currently known whether the observed increase in the likelihood of dysphagia occurring following subcortical lesions is due to damage to subcortical areas or disruption to the connections between cortical and subcortical structures.”

Line 375: It has been at least postulated that the reason for haemorrhagic strokes resulting in dysphagia more frequently than ischaemic strokes is that they result in larger lesions (hence greater impairment)

Thank you. The text has been changed to:

“The precise reason for this is currently unknown but it is currently thought that this observation is related to the increased size of haemorrhagic compared to ischaemic lesions (Jørgensen et al., 1995). The larger the lesion, the greater its deleterious effect hence the higher National Institutes of Health Stroke Scale (NIHSS) scores in patients with haemorrhagic compared to ischaemic strokes as well as their longer hospital length of stays and increased mortality (Vasan et al., 2023).”

In reference to Line 389 "neuroplastic compensation is the process that underlies the recovery of swallowing", this statement does not account for recovery / repair in the area of the lesion (which is supported in the following paragraph referencing knowledge from animal model lesion and recovery studies). Furthermore, it is a rather strong statement being that it is based off of a study with only 11 patients with post-stroke dysphagia at the three month recovery point. It is suggested to rephrase to better represent the state of the evidence.

Thank you. This has been changed to:

“Post lesional neuroplastic changes in damaged and undamaged areas of the brain(Hamdy et al., 1998) are thought to be the underlying reason for the recovery of swallowing function. Functional imaging studies have shown ipsilesional brain changes occurring in patients with PSD (Li et al., 2009; Malandraki et al., 2011) and limb dysfunction (Nelles et al., 1999). However, specifically regarding recovery in patients with PSD, the contralesional sensorimotor cortex has been observed to exhibit a greater increase in activity than was observed over the affected hemisphere (Hamdy et al., 1998). This observation was thought to be compensation for the damage caused to its hemispheric partner. This contralesional compensatory process was first shown in 1998 by Hamdy et al and has also been

observed to occur in subsequent PSD functional imaging studies (Almeida et al., 2017; Li et al., 2009; Mihai et al., 2016)."

Line 446: This phrase "rehab only provides modest improvements to swallow safety" is supported by only one study and does not accurately characterize the impact of rehabilitation on countless patients with post-stroke dysphagia. It is erroneous, and frankly dangerous, to suggest as a blanket statement that rehab does not help. While it might not help all recover to be able to eat fully by mouth, especially those with very severe conditions, it does remarkably change the lives of others. I strongly suggest that this statement be modified to reflect that with improved knowledge to harness neuroplasticity that we could optimize rehabilitation results for all with post-stroke dysphagia and to help those with very severe conditions that are currently unable to eat by mouth. (Note: please be careful with these types of erroneous blanket statements, in the U.S. insurance providers will use this to start denying rehabilitation access to patients with post-stroke dysphagia. We do so much good to help so many patients, it would be truly horrible if that happened.)

Thank you. The text has been changed to:

"Rehabilitation involves patients being taught exercises or manoeuvres to improve the efficiency and safety of their swallowing (I. Cheng, A. Sasegbon, et al., 2022) (Robbins et al., 2007). Rehabilitative interventions have been shown to lead to significant improvements in oropharyngeal muscle strength and bulk (Park et al., 2019). They have been used for decades as a means to improve quality of life in patients with PSD and currently constitute the cornerstone of neurogenic dysphagia treatment in clinical settings."

And

"Furthermore, no studies have investigated the effect on dysphagia of rehabilitative exercises used alongside novel neuromodulatory interventions. These large gaps in our understanding will need to be addressed by future studies."

Line 453: This statement misses that a fundamental goal of rehabilitation exercises is to modify the neurophysiology of swallowing. It is hard to imagine how exercises to improve swallowing, that are sufficiently effective, wouldn't result in neurological / cortical changes. Similarly, the goal of neurophysiological changes due to neuromodulation without resulting changes in swallowing function would not be relevant. Please nuance this statement.

Thank you. The text now reads:

"Functional imaging studies have also shown increased brain activity in swallowing areas following rehabilitative exercises (Peck et al., 2010). Hence, these exercises act to both improve oropharyngeal muscle strength but also modulate the neuronal control of those strengthened muscles. However, it should be noted that despite rehabilitative exercises forming a key part of the treatment armamentarium in clinical practice, there are few studies that have sought to investigate the impact of different rehabilitative exercises on swallowing safety."

Lines 508-509: Related to the statement that meta-analyses have demonstrate that all these neurostim techniques are effective at improving PSD in the short and medium term, there are two meta-analyses provided to support this. Both meta-analyses are based on the rather limited (in terms of sample size for the individual studies) data. All of the studies included in the meta-analyses use very different neurostimulation techniques and methods as well as different outcome variables. Given the state of the currently available evidence that precludes the conduct of a meta-analysis combining studies of similar methods and similar outcomes, it is suggested that this statement be nuanced to read that there is emerging evidence that neurostimulation techniques hold promise for improving PSD.

Thank you. The text has been changed to:

“Recent meta-analyses show that there is some evidence that these techniques are effective over the short and medium term at improving PSD (Cheng et al., 2020; Zhu & Gu, 2022) with contralesional and bilateral stimulation appearing to be more effective than ipsilesional stimulation in PSD (Cheng et al., 2020). However, it must be noted that due to the comparatively limited number of studies in the field, the studies incorporated into the metanalyses were heterogeneous with respect to their treatment protocols and outcome measures.”

REQUIRED ITEMS

- Please include an Abstract Figure file, as well as the Figure Legend text within the main article file. The Abstract Figure is a piece of artwork designed to give readers an immediate understanding of the Review Article and should summarise the main conclusions. If possible, the image should be easily 'readable' from left to right or top to bottom. It should show the physiological relevance of the Review so readers can assess the importance and content of the article. Abstract Figures should not merely recapitulate other figures in the Review. Please try to keep the diagram as simple as possible and without superfluous information that may distract from the main conclusion of the Review. Abstract Figures must be provided by authors no later than the revised manuscript stage and should be uploaded as a separate file during online submission labelled as File Type 'Abstract Figure'. Please ensure that you include the figure legend in the main article file. All Abstract Figures will be sent to a professional illustrator for redrawing and you may be asked to approve the redrawn figure before your paper is accepted.

This has been done.

- The reference list must be in alphabetical order, rather than numbered, to comply with our [Journal format \[jp.msubmit.net\]](http://jp.msubmit.net).

Thank you this has been done.

- Please upload separate high quality figure files via the submission form.

This has been done.

Dear Professor Hamdy,

Re: JP-TR-2024-285564R1 "The neurorehabilitation of post-stroke dysphagia - physiology and pathophysiology" by Ayodele Sasegbon, Ivy Cheng, and Shaheen Hamdy

We are pleased to tell you that your paper has been accepted for publication in The Journal of Physiology.

*****IMPORTANT*****

There is one point that we need to check with you before sending to production - the Abstract Figure, and Figure 1, appear to be identical? Is that correct?

If so, we would recommend that Figure 1 is removed (to avoid duplication) and the Abstract Figure is retained. Current Figure 2 would then become Figure 1.

Can you confirm this is OK, please? We can make the required amendments at this end for you.

Thank you - we look forward to hearing from you.

Authors should note that it is too late at this point to offer corrections prior to proofing. Major corrections at proof stage, such as changes to figures, will be referred to the Editors for approval before they can be incorporated. Only minor changes, such as to style and consistency, should be made at proof stage. Changes that need to be made after proof stage will usually require a formal correction notice.

Yours sincerely,

Laura Bennet
Senior Editor
The Journal of Physiology

P.S. - You can help your research get the attention it deserves! Check out Wiley's free Promotion Guide for best-practice recommendations for promoting your work at www.wileyauthors.com/eeo/guide. You can learn more about Wiley Editing Services which offers professional video, design, and writing services to create shareable video abstracts, infographics, conference posters, lay summaries, and research news stories for your research at www.wileyauthors.com/eeo/promotion.

IMPORTANT NOTICE ABOUT OPEN ACCESS: To assist authors whose funding agencies mandate public access to published research findings sooner than 12 months after publication, The Journal of Physiology allows authors to pay an Open Access (OA) fee to have their papers made freely available immediately on publication.

You can check if your funder or institution has a Wiley Open Access Account here: <https://authorservices.wiley.com/author-resources/Journal-Authors/licensing-and-open-access/open-access/author-compliance-tool.html>.

EDITOR COMMENTS

Reviewing Editor:

Thank you for responding in a comprehensive fashion to the detailed comments provided by the referee, I share the view of the referee that the paper has the potential to make a significant impact upon the field.

REFeree COMMENTS

Referee #1:

All prior comments have been addressed. It is expected that this manuscript will be a considerable contribution to the field.
Thank you!

1st Confidential Review

23-Feb-2024